# The Effect of Sex and Different Repetition Maximums on Kinematics and Surface Electromyography in the Last Repetition of the Barbell Back Squat

**DOI:** 10.3390/jfmk9020075

**Published:** 2024-04-18

**Authors:** Andreas Hegdahl Gundersen, Hallvard Nygaard Falch, Andrea Bao Fredriksen, Roland van den Tillaar

**Affiliations:** Department of Sports Sciences, Nord University, 7600 Levanger, Norway; andreasgundersen78@gmail.com (A.H.G.); falch7@hotmail.com (H.N.F.); andreabaof@hotmail.com (A.B.F.)

**Keywords:** resistance, angular velocity, strength, sticking region

## Abstract

During the ascent phase of a maximal barbell back squat after an initial acceleration, a deceleration region occurs as the result of different biomechanical factors. This is known as the sticking region. However, whether this region is similar in the last repetition of different repetition maximums and if sex has an impact on biomechanics of this region are not known. Therefore, this study investigated the effect of sex (men/women) and repetition maximum (1-, 3-, 6-, and 10RM) on kinematics and surface electromyography around the sticking region. Twenty-six resistance-trained individuals comprising 13 men (body mass: 82.2 ± 8.7; age: 23.6 ± 1.9; height: 181.1 ± 6.5) and 13 women (body mass: 63.6 ± 6.6; age: 23.9 ± 4.5; height: 166.0 ± 4.5) participated in the study. The main findings were that women, in comparison to men, displayed larger trunk lean and lower hip extension angles in the sticking region, possibly due to different hip/knee extensor strength ratios. Moreover, an inverse relationship was discovered between repetition range and timing from V_0_ to V_max2_, in which lower repetition ranges (1- and 3RM) were shorter in V_max2_ compared to higher ranges (6- and 10RM). It was concluded that this occurrence is due to more moments of inertia in lower repetition ranges. Our findings suggest that both sex and repetition range might induce different requirements during the squat ascent.

## 1. Introduction

The barbell back squat (squat) is a multi-joint resistance exercise incorporated into training programmes by a wide variety of cohorts, with the aim of enhancing everything from rehabilitation and health benefits to performance in sports, through increasing power, strength, and hypertrophy in the lower extremities [1]. Several studies have investigated biomechanical limiting factors in the squat among strength-trained individuals [2,3,4,5,6,7,8]. Several of those studies have separated the squat ascent into three regions: pre-sticking region (bottom position: V_0_—first peak velocity: V_max1_), sticking region (V_max1_—minimum velocity: V_min_), and post-sticking region (V_min_—second peak velocity: V_max2_) [5,6,7]. Larsen, et al. [9] observed that as loads increased, the sticking region occurred at lower barbell heights with lower hip and knee extension angles. The authors speculated that the lower sticking region observed with increased barbell load resulted in a disadvantageous internal moment arm position for the hip and knee extensors, as the gluteus maximus and vasti muscles have been reported to decrease their internal moment arm with increased flexion angles [10,11]. Moreover, two studies also reported a greater forward trunk lean when sets were taken within 80% of one repetition maximum (1RM), creating a larger moment arm and therefore increased rotational work at the hip [4,9].

The measurement of 1RM is a popular way of testing maximal strength among multiple different athletic cohorts because maximum external barbell load lifted for one repetition is representative of an athlete’s ability to exert force [12]. Strength and hypertrophic responses are known to occur at multiple different repetition intervals [13]. The current consensus is that strength adaptions are maximised when training with heavier loads and lower per-set repetitions, while hypertrophic adaptions transpire on a larger spectrum of per-set repetitions but require proximity to muscular failure [13]. Therefore, training both strength and hypertrophic specifically should result in a sticking region in the final repetitions, as the literature suggests a sticking region to occur in lifts corresponding to >80% of 1RM [5,6]. However, none of the aforementioned studies have compared biomechanics around the sticking region directly between different RMs in the squat. In bench press, in the last repetition of 1-, 3-, 6-, and 10RM, kinematics and surface electromyography (sEMG) were compared [14], finding mostly similar sEMG and joint kinematics between the conditions, but higher barbell velocity across two events in 10RM compared to 1RM. The extension of these findings to the squat is currently unknown, as the squat engages larger muscle groups, which is known to impose heightened metabolic demands [15]. However, based upon Henneman’s size principle, muscular excitation should remain similar across different RMs considering their proximity to failure is equal, as the principle relates the basis for size-ordered activation of motor units [16].

Biomechanical variations may be complicated further when accounting for differences between the sexes. Men are found to produce more force per unit of mass, which is on a populational average due to more muscle mass and less fat mass, thus expressing greater absolute and relative strength [17,18,19]. Additionally, greater sex differences in lean upper body mass compared to lean lower body mass were found [17,18,19]. When normalised for body mass, studies consistently find men to generate greater knee extensor torque than women during maximal effort contractions [20,21,22,23], which also seems to apply for the hip extensors [24,25,26]. In addition, women possess a knee/hip strength ratio close to 1, suggesting equal strength in knee and hip extensors [24]. Conversely, men had a ratio <1, which implies the hip extensors to be stronger relative to the knee extensors [24]. Furthermore, women have been shown to possess heightened strength endurance capacities at given relative loads [17,27,28]. This occurrence is postulated to arise from an augmented proportion of type 1 muscle fibres [17], better reliance on fat oxidation [27], lower oxygen demands attributable to lower muscle mass [27], and reduced work per repetition due to shorter limbs [29]. The precise mechanistic underpinnings of the observed disparities in strength endurance remain uncertain, with speculation that such variations may be reliant upon the specific strength task performed, as sEMG has been found to be similar between the sexes when normalised for strength [30]. Although several mechanisms could cause exercise form breakdown [9], similar sEMG and kinematics when comparing men and women in squats of different RM could indicate similar requirements for lifting through the sticking region and completing the squat ascent.

Therefore, the purpose of this study was to investigate the impact of four repetition maximums (1-, 3-, 6-, and 10RM) and sex on kinematics and sEMG amplitude of the last repetition in the squat. When viewed synergistically, this could provide insight on form breakdown as sets of different loads reach maximum. Such information could offer useful inputs in terms of training specificity when individualising training programs for strength and/or hypertrophy in the squat. Based on Larsen, Kristiansen, Nygaard Falch, Estifanos Haugen, Fimland and van den Tillaar [9], the Henneman’s size principle, and Nimphius, et al. [30], it was hypothesised that no difference in sEMG would occur between neither RM nor sex [16]. Lastly, no kinematic differences were hypothesised between sexes, but based on Larsen, Kristiansen, Nygaard Falch, Estifanos Haugen, Fimland and van den Tillaar [9], the timing of different events was expected to be longer in the higher load (1- and 3RM) sets compared to lower load (6- and 10RM) sets.

## 2. Materials and Methods

### 2.1. Participants

A total of 26 recreationally strength-trained participants comprising both men (*n* = 13) and women (*n* = 13) volunteered to partake in the study (Table 1). Inclusion criteria stipulated that participants had to manage a squat equivalent to 1.2 × body mass (men) and 1 × body mass (women), adhering to the technique requirements established by the International Powerlifting Federation, which requires the femur to be parallel to the floor at bottom position. Additionally, participants had to declare absence of any injury or illness which could impede maximum effort. Furthermore, participants were instructed not to engage in any lower limb exercise and refrain from alcohol >48 h prior to testing. The risk and benefits of participation were explained both in writing and orally, and written consent had to be signed before participation. The study was approved by the local ethics committee and the Norwegian Centre for Research Data (project no. 701688), in conjunction with the latest alteration of the Helsinki Declaration.

### 2.2. Procedure

To investigate the potential impact of sex and RM (1-, 3-, 6-, and 10RM) on kinematics and sEMG amplitude around the sticking region in the final repetition, a randomised mixed repeated-measures design was assessed.

The familiarisation test mirrored the same test protocol as the experimental test, serving to establish the appropriate load for each repetition range, whereas kinematic and sEMG data were collected solely during the experimental test session. To enhance ecological validity and reliability, stance width was standardised to the personal preference of each participant. The use of lifting aids (e.g., knee sleeves, lifting belt) was prohibited, with the exception of lifting shoes. Both sessions commenced with a standardised warm-up protocol squatting at incrementally higher percentages of estimated 1RM (40-, 60, 70, and 80%), before squatting 1-, 3-, 6-, and 10RM in a randomised order (Figure 1) decided by an online randomiser (https://www.random.org, accessed on 26 March 2024). The subjects were not restricted in lifting tempo and used their self-selected tempo, but were not allowed to remain at lockout for longer than 2 s. To minimise the risk of fatigue-induced performance, each participant was required to rest for a minimum of five minutes between each set for both test sessions. The familiarisation test started by acquiring the participant’s preferred stance width and body height using a measuring tape, before body mass was measured with a standing scale (Soehnle Professional 7830, standing scale). Squatting depth at the bottom of the descending phase was defined in accordance with the technique regulations set by the International Powerlifting Federation, which necessitates the trochanter major to be inferior to the patella. To ensure reliability, appropriate squatting depth was monitored using a three-dimensional motion capture system.

For analysis, the squat ascent was separated into four events (bottom position: V_0_, first peak velocity: V_max1_, minimum velocity: V_min_, and second peak velocity: V_max2_), which divided the ascent into three different phases (V_0_–V_max1_: pre-sticking region, V_max1_–V_min_: sticking region, and V_min_–V_max2_: post-sticking region, Figure 1) [6].

sEMG data were recorded and analysed using Musclelab v.10.200.90.5095 (Ergotest Technology, Langesund, Norway). Electrodes sampling at 1000 Hz (Zynex Neurodiagnostics, Lone Tree, CO, USA) were lubricated and affixed lengthwise in the presumed direction of the underlying muscle fibre to the dominant side of 10 different muscles (erector spinae iliocostalis, gluteus maximus, gluteus medius, semitendinosus, bicep femoris, vastus lateralis, vastus medialis, rectus femoris, gastrocnemius, and soleus). The placement of the different electrodes was conducted according to SENIAM recommendations [31]. Prior to attachment, each participant underwent appropriate shaving and cleansing with alcohol to reduce interference from hair and dead skin. The root mean square of the unprocessed sEMG signal during the three regions (pre-sticking, sticking, and post-sticking) was computed by a hardware circuit network (frequency response 20–500 Hz, with a moving average filter of 100 ms width, securing an overall error rate of ±0.5%). sEMG was normalised by each individual peak sEMG amplitude during one of the regions of the last repetition of a repetition maximum and defined as 100%. A linear encoder sampling at 200 Hz (ET-Enc-02, Ergotest Technology AS, Langesund, Norway) was attached to the barbell and used to synchronise sEMG data and kinematics data to events and regions.

Eight three-dimensional motion capture cameras (Qualisys, Gothenburg, Sweden) operating at a frequency of 500 Hz were used to track reflective markers for events V_0_, V_max1_, V_min_, and V_max2_. The reflective markers (14 mm) were placed on anatomical landmarks on both sides of the body (acromion, pelvis, iliac crest, posterior superior iliac spine, trochanter major, the medial and lateral condyle of the knee, medial and lateral malleolus, sternum, tuber calcanei, and 1st and 5th proximal phalanx), creating a three-dimensional measurement of each participant. This facilitated the determination of sagittal-plane kinematics for the hip, knee, and ankle joints, which in an erect standing position were defined as 180° for hip and knee, 90° for the ankle, and 0° for the trunk. Timing of the different events and peak joint angular velocities was calculated during the ascending phase. All kinematic data were transported via C3D files to Visual3D (C-motion, Germantown, MD, USA) for segment building and subsequent analysis. All joint angles were defined as the proximal segment relative to the distal segment, except the trunk angle which was defined as the trunk relative to the laboratory floor.

### 2.3. Statistical Analysis

Data were expressed as means and standard deviations (SD) per sex and normality of data were assessed and confirmed using the Shapiro–Wilk test. Differences in anthropometrics and lifted load at the different repetition maximums between the sexes were assessed with independent sample t-tests. To compare sex differences in joint kinematics, peak/minimum angular velocity, and timing across different repetition ranges (1-, 3-, 6-, and 10RM), a 2 (sex: men/women; independent measures) by 4 (event: V_0_, V_max1_, V_min_, and V_max2_; repeated measures) analysis of variance (ANOVA) was assessed. To compare sEMG amplitude between sexes, a mixed 2 (sex: men/women) by 4 (repetition range: 1-, 3-, 6-, and 10RM) by 3 (region: pre-sticking, sticking, and post-sticking) with repeated measures was assessed to compare each muscle. Post hoc comparison with a Holm–Bonferroni correction was conducted when significant differences were observed. The assumption of sphericity was controlled with Mauchly´s test of sphericity. If the assumption of sphericity was violated, the Greenhouse–Geisser adjusted *p*-value was reported. The level of significance was set at *p* < 0.05. Data are reported as means ± standard deviations. Effect size was evaluated as eta partial squared (η_p_^2^), whereby 0.01 to 0.06 η_p_^2^ constitutes a small effect, 0.06 to 0.14 was defined as a medium effect, and 0.14 > η_p_^2^ was defined as a large effect [32]. The statistical analysis was conducted in IBM SPSS Statistics 27.0 (IBM, Armonk, NY, USA).

## 3. Results

Women were significantly lighter, shorter, and had a lower absolute and relative strength across all repetition ranges (Table 1 and Table 2).

Significant sex differences were observed in hip extension angles at V_min_ and V_max2_, and in trunk lean at V_max1_, V_min_, and V_max2_ (F ≥ 5.834; *p* ≤ 0.026; η^2^ ≥ 0.235). Post hoc testing revealed that the hip extension angle in men was significantly greater across all repetition ranges in V_min_ and V_max2_ when compared to women. Trunk lean angle in women was significantly higher across all repetition ranges at V_max1_ and V_min_ in comparison to men (Figure 2A). No significant sex differences were found in knee extension angle and ankle plantar flexion angle (F ≤ 2.355; *p* ≥ 0.14; η^2^ ≤ 0.101, Figure 2C,D).

A significant effect of repetition ranges (men and women together) was observed in both knee extension angle and ankle plantar flexion angle at V_min_ (F ≥ 3.689; *p* ≤ 0.041; η^2^ ≥ 0.154). Post hoc tests revealed that the knee extension angle and ankle plantar flexion were significantly greater in 6RM compared to 1RM at V_min_ (Figure 2C,D). No significant effect of repetition ranges was found in hip extension angle and trunk lean angle (F ≤ 1.614; *p* ≥ 0.196; η^2^ ≤ 0.078, Figure 2A,B). Furthermore, a significant interaction between repetition range and sex was observed in the knee extension angle at V_max2_ (F ≥ 5.415; *p* ≤ 0.021; η^2^ ≥ 0.253, Figure 2C).

A significant effect of repetition range on peak knee angular velocity was found (F = 3.38; *p* = 0.025; η^2^ = 0.158). The post hoc test revealed the peak knee angular velocity to be significantly higher in 3- and 6RM compared to 10RM (Figure 3). No other significant effect of repetition range on peak angular velocity was observed in any of the other joints (F ≤ 2.044; *p* ≥ 0.119; η^2^ ≤ 0.107). Moreover, no significant differences between sexes (F ≤ 0.37; *p* ≥ 0.08; η^2^ ≤ 0.13, Figure 3) or interaction effects for peak angular velocity were discovered (F ≤ 1.41; *p* ≥ 0.247; η^2^ ≤ 0.06, Figure 3).

A significant effect of repetition range was found on the timing of V_max2_ (F = 9.243; *p* < 0.001; η^2^ = 0.327), in which the timing of V_max2_ in 10RM was significantly shorter compared to 1- and 3RM. In addition, the timing of V_max2_ in 6RM was shorter in comparison to 1RM. No significant effects of repetition range were observed in the timing of V_max1_ and V_min_ (F ≤ 3.176; *p* ≥ 0.07; η^2^ ≤ 0.137). Lastly, no significant sex differences were discovered (F ≤ 1.085; *p* ≥ 0.311; η^2^ ≤ 0.054, Figure 4).

A significant effect of repetition range was observed in sEMG amplitude for the vastus medialis, vastus lateralis, soleus, and gastrocnemius in the post-sticking region, and for the vastus lateralis in the sticking region (F ≥ 4.019; *p* ≤ 0.011; η^2^ ≥ 0.041, Figure 5D). Post hoc analysis revealed significantly higher sEMG amplitude when comparing 1RM with 10RM at the post-sticking region in the vastus medialis, vastus lateralis, soleus, and gastrocnemius. Additionally, the vastus lateralis showed higher sEMG amplitude when comparing 3RM with 6- and 10RM in the sticking region, and when comparing 3RM with 10RM at the post-sticking region. No significant effect of repetition range was observed in any of the other muscles (F ≤ 2.484; *p* ≥ 0.071; η^2^ ≤ 0.020, Figure 5). Also, a significant effect of region was discovered in all muscles (F ≥ 6.382; *p* ≤ 0.012; η^2^ ≥ 0.148), except at the erector spinae (F = 2.676; *p* = 0.083; η^2^ = 0024, Figure 5B). In addition, significant interactions between sex and repetition range were found in the vastus lateralis and gluteus medius (F ≥ 3.168; *p* ≤ 0.049; η^2^ ≥ 0.023, Figure 5B,E). Moreover, significant interactions between sex and region were discovered at the soleus (F ≥ 3.980; *p* ≤ 0.0027; η^2^ ≥ 0.030, Figure 5I). Lastly, a significant interaction between region, repetition range, and sex was observed for both the gluteus medius and gluteus maximus (F ≥ 2.341; *p* ≤ 0.036; η^2^ ≥ 0.005, Figure 5E,F). No significant interactions were found in any of the other muscles (F ≤ 2.748; *p* ≥ 0.078; η^2^ ≤ 0.020, Figure 5).

## 4. Discussion

The aim of this study was to investigate the effect of different RM ranges (1-, 3-, 6-, and 10RM) and sex on kinematics and sEMG amplitude around the sticking region in the final repetition of the squat. The main findings were that in all repetition ranges, women displayed a larger trunk lean and lower hip extension angle in the sticking region when compared to men. Furthermore, an inverse relationship between repetition range and timing from V_0_ to V_max2_ was found, such that timing was shorter at 6- and 10RM compared to 1RM, and shorter at 10RM compared to 3RM. Lastly, the lower repetition range (1- and 3RM) displayed higher sEMG amplitude than the higher repetition range (6- and 10RM).

An inverse relationship between repetition range and timing from V_0_ to V_max2_ was discovered, in that such timing was shorter for the higher repetition range (6- and 10RM) compared to the lower repetition range (1- and 3RM). From a biomechanical perspective, this was expected, as per-set repetitions and barbell load are inversely related; hence, lower per-set repetition ranges should be influenced by more moments of inertia, resulting in a slower ascent phase. This was in accordance with what was reported in bench press activity: significantly higher peak barbell velocities in the sticking region of 10RM compared to 1RM [14]. Conveniently, we observed a difference in timing to V_max2_, which is the event subsequent to the sticking region. As such, when seen together with the findings from Larsen, Haugen and van den Tillaar [14], a tendency of velocity discrepancy between loads might exist in the sticking region. This is logical, as more inertia adds torque to the already heightened rotational work at the hip during the sticking region [3], which slows down the hip extension to keep the net hip moment similar.

Women displayed a larger trunk lean and lower hip extension angle in the sticking region in all repetition ranges when compared to men, which indicates technique differences between the sexes (Figure 2 and Figure 6).

When normalised for body mass, studies consistently find men to generate greater knee and hip extensor torque than women during maximal effort contractions [20,21,22,23,24,25,26]. Stearns, Keim and Powers [24] found that the hip extensors of men were 44% stronger than the hip extensors of women, whereas the knee extensors of men were only 28% stronger than those of women (24). Additionally, women were found to possess a hip/knee extensor strength ratio close to 1, indicating parity in strength across knee and hip extensors. Conversely, men demonstrated a ratio <1, signifying a relative strength dominance in hip extensors compared to the knee extensors [24]. As reported by Larsen, Kristiansen and van den Tillaar [7], the knee moment contribution decreases in the sticking region, while hip moment arm and hip moment contribution increase. Also, the gluteus maximus has been found to be at a mechanical disadvantage to exert force in the sticking region [33]. Thus, when hip moment contribution increases in the sticking region, hip extensors have been viewed as a bottleneck in maximal squats [3]. Accordingly, the biomechanical requirements of the sticking region might disproportionally increase difficulty for lifters with weaker hip extensors relative to knee extensors. Therefore, as the hip joint is responsible for extending the hip, and thereby the trunk, slower extension of the hips in the sticking region could explain why the women of this study squat with increased hip extension angles and trunk lean in the sticking region. Even with higher peak angular hip and trunk extension velocities (Figure 2) of women (no significant moderate effect size), due to lower angles during the sticking region compared to men, these could only partly compensate for this later in the lift, as shown by similar trunk lean at V_max2_ (Figure 2A). The hip extension angle is still significantly higher in men, probably because the peak hip extension occurs at around V_max2_ [34]. However, more research regarding kinematic discrepancies between the sexes in maximal squats must be conducted in order to draw more concise conclusions.

There were no significant sEMG findings supporting these kinematic sex differences, as both hip and knee extensors showed similar sEMG. However, the study used a mixed design, so sEMG was normalised, with each individual’s peak sEMG amplitude defined as 100%. As such, comparing peak values between groups is not possible. Consequently, comparable sEMG only indicates that the timing of excitation is similar between sexes in the different regions.

It was hypothesised, likewise, that no differences would be found in sEMG activity between the repetition ranges, as the recruitment of motor units is size ordered. Hence, the highest threshold motor units would be recruited when the sets reached failure, creating similar sEMG amplitudes [16]. However, knee extensors (medial and lateral vastus) and plantar flexors displayed a lower sEMG amplitude at the post-sticking region in 10RM compared to 1 RM, with the vastus lateralis also reporting a lower sEMG amplitude in 6- and 10RM compared to 3RM at the sticking and post-sticking regions. Using a fatigue-inducing protocol, Tesch, et al. [35] found a parallel reduction in sEMG amplitude and maximum force in the knee extensors as the accumulation of lactic acid increased, possibly due to changes in action potential in the most acid-labile motor units. Exercise on multiple large muscle groups is known to increase acute metabolic demands [15], such that when combined with longer work duration in 10RM (>15 s) compared to 1 RM (<5 s), this may have caused higher threshold motor units to fail due to acidosis, which, in turn, reduced sEMG amplitude in 10RM. Additionally, previous research has observed reduced knee flexion and increased hip flexion as a high-repetition squat nears failure, making the lift increasingly hip dominant [36]. As such, knee extensors and ankle plantar flexors may have experienced heightened ATP depletion during initial repetitions in 10RM conditions [37], as the squat can be performed with a more knee-dominant technique when perceived effort is low (which also augments ankle dorsal flexion) [38]. Hence, recruitment of motor units and therefore sEMG amplitude in 10RM may be modulated by locally fatiguing factors, including ATP depletion and decreased oxidation associated with longer set duration [39,40,41].

This study has limitations that should be addressed. Firstly, despite the participants engaging in sets with varying loads, the consideration of body mass above the knees was omitted, which holds significance because it contributes to the absolute weight lifted. Secondly, bar placement was not standardised among participants, so we cannot exclude the possible influence of high-bar vs. low-bar technique. Thirdly, lifting tempo was not controlled for, which could influence the results. However, to avoid extra constraints on the subjects, which would make this study less ecological, it was decided that the lifting tempo and barbell placement between subjects was not standardised.

## 5. Conclusions

This study revealed notable variations in squatting technique between the sexes across all RMs (1-, 3-, 6-, and 10RM), with women exhibiting greater trunk lean and reduced hip extension angles compared to men in and around the sticking region, possibly due to differences in hip/knee extensor strength ratios. Additionally, an inverse relationship between timing to V_max2_ and repetition range was discovered, where timing was lower for lower repetition ranges (1–3RM) compared to higher repetition ranges (6–10RM), which might occur as heavier loads are influenced by more moments of inertia. Therefore, based upon the principle of specificity, sex and repetition range might induce different requirements during the ascent phase when squatting to volitional failure.

## Figures and Tables

**Figure 1 jfmk-09-00075-f001:**
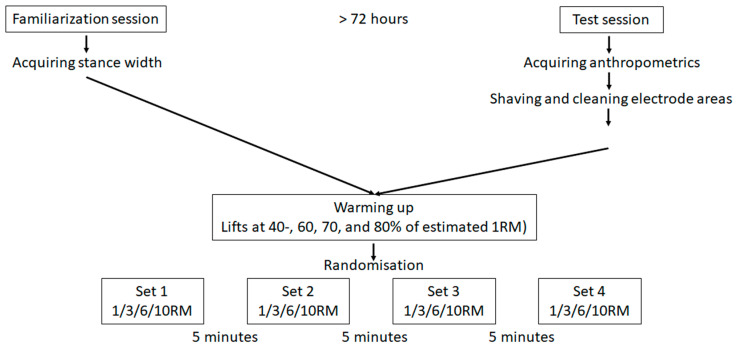
Test protocol.

**Figure 2 jfmk-09-00075-f002:**
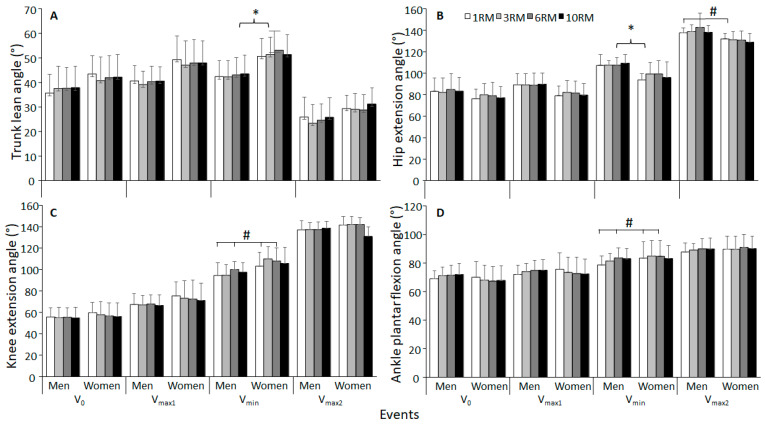
Mean ± SD (**A**) trunk, (**B**) hip, (**C**) knee, and (**D**) ankle angle at the different events of the last repetition during 1-, 3-, 6-, and 10RM barbell back squat. * Significantly different joint angles between sexes for all repetition ranges (*p* < 0.05). # Significantly different joint angles between two repetition ranges for both sexes (*p* < 0.05).

**Figure 3 jfmk-09-00075-f003:**
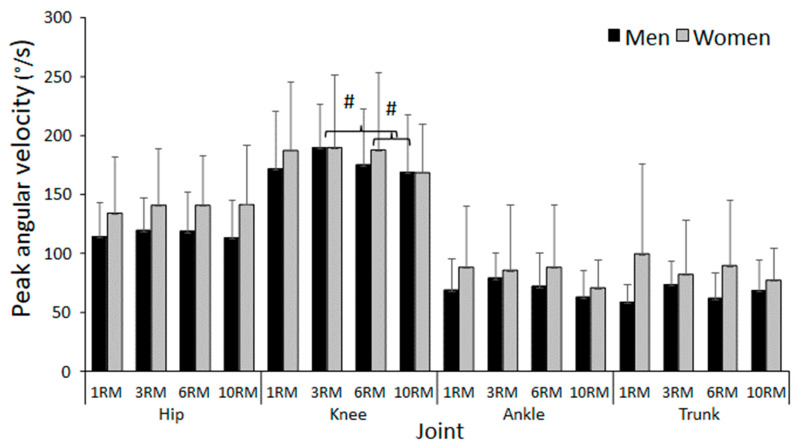
Mean ± SD peak angular velocity at the different events of the last repetition during 1-, 3-, 6-, and 10RM barbell back squat. # Significantly different peak angular velocity between these two repetition ranges for this joint (*p* < 0.05).

**Figure 4 jfmk-09-00075-f004:**
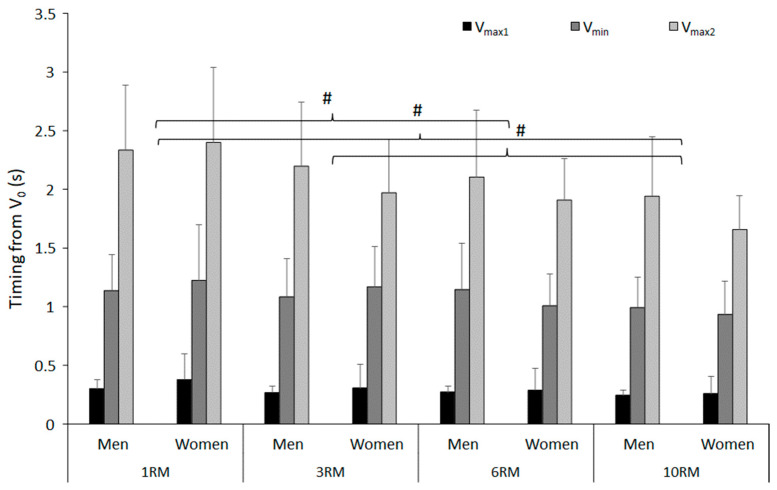
Mean ± SD of the timing from V_0_ to the different events at the last repetition during 1-, 3-, 6-, and 10RM barbell back squat. # Significantly different timing of V_max2_ between these two repetition ranges for both sexes (*p* < 0.05).

**Figure 5 jfmk-09-00075-f005:**
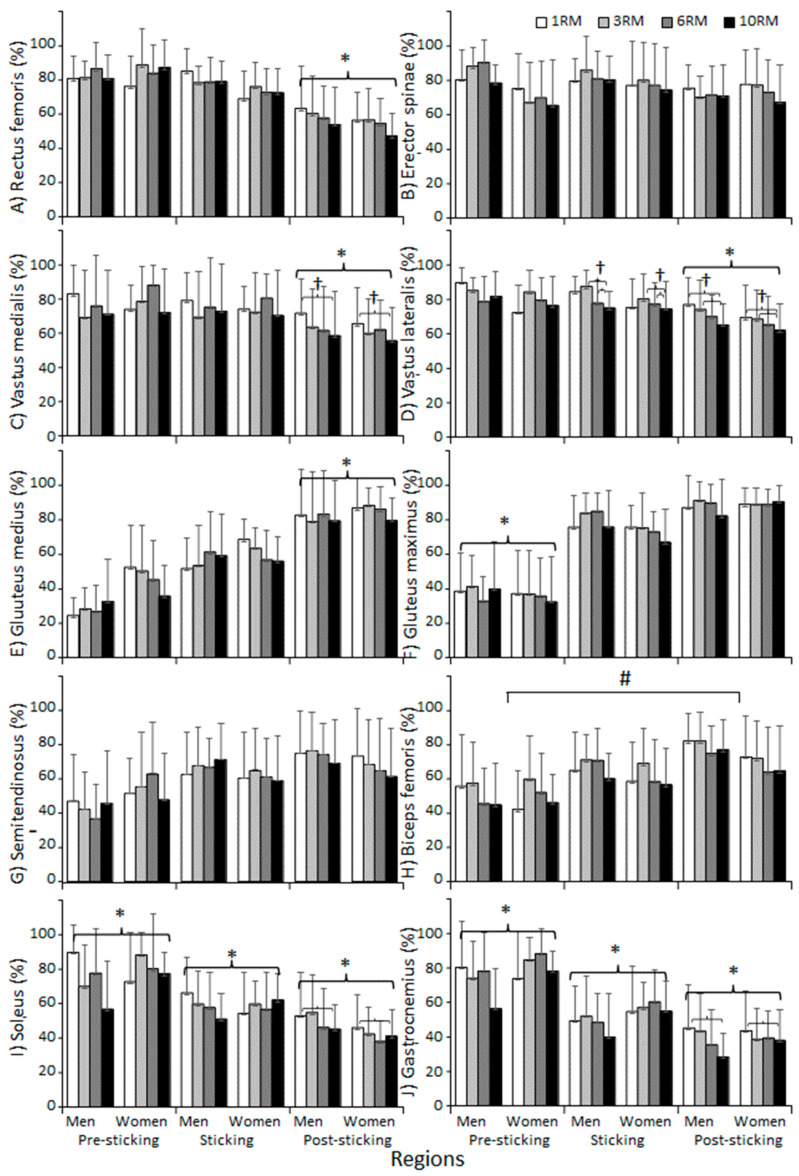
Mean ± SD of sEMG of the different muscles (**A**–**J**) during the different phases of the last repetition during 1-, 3-, 6-, and 10RM barbell back squat. * Significantly different sEMG for all repetition ranges for this region with all other regions (*p* < 0.05). # Significantly different sEMG for both men and women between these two regions for all repetition ranges (*p* < 0.05). † Significantly different sEMG between these two repetition ranges for this region (*p* < 0.05).

**Figure 6 jfmk-09-00075-f006:**
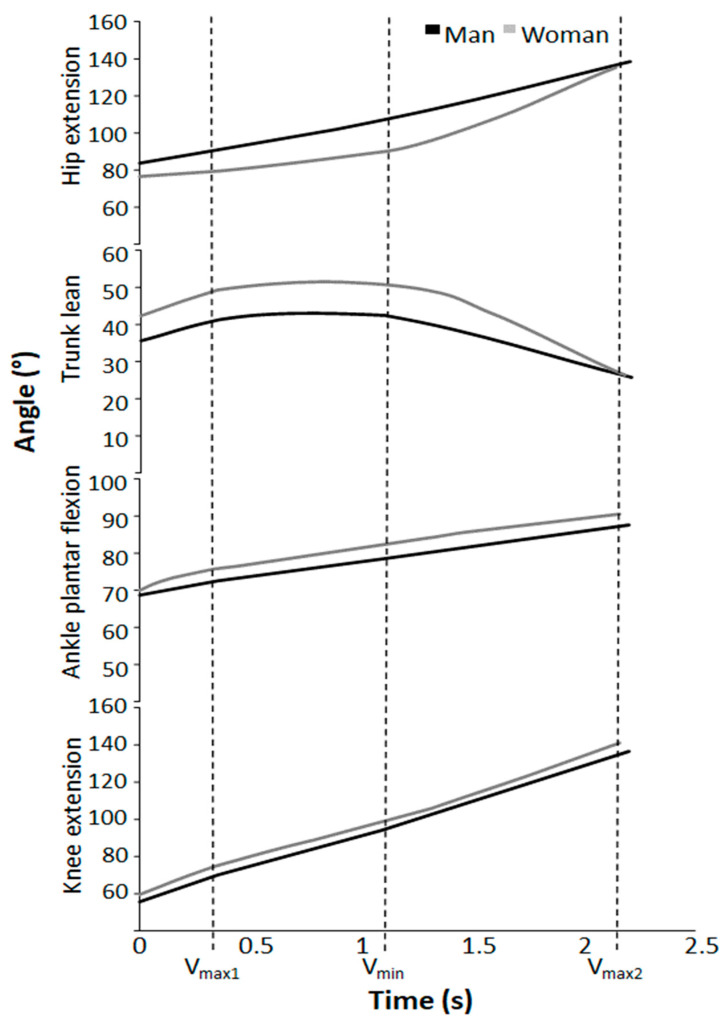
Two examples of joint angles over time during squats for a representative man and woman.

**Table 1 jfmk-09-00075-t001:** Mean age (years), height (cm), and body mass (kg) for men and women.

Sex	Age (Years)	Height (cm)	Body Mass (kg)
Men	23.6 ± 1.9	181.1 ± 6.5	82.2 ± 8.7
Women	23.9 ± 4.5	166.0 ± 4.5	63.6 ± 6.6

**Table 2 jfmk-09-00075-t002:** Mean weights (kg) lifted for the different repetition ranges and mean relative strength for men and women.

Sex	1RM (kg)	3RM (kg)	6RM (kg)	10RM (kg)	Relative Strength
Men	100.3 ± 26.9	90.2 ± 25.0	81.8 ± 22.4	74.4 ± 21.4	1.3 ± 0.2
Women	77.8 ± 11.9	69.1 ± 11.3	62.5 ± 8.2	56.6 ± 8.2	1.2 ± 0.1

## Data Availability

The data presented in this study are available on request from the corresponding author. The data are not publicly available due to national laws of the Norwegian government on privacy.

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
