# Peer review of "The Effect of Sex and Different Repetition Maximums on Kinematics and Surface Electromyography in the Last Repetition of the Barbell Back Squat"

_jfmk, 2024, doi:10.3390/jfmk9020075_

Round 1
Reviewer 1 Report (Previous Reviewer 2)
Comments and Suggestions for Authors
Thank you for your hard work on the revision.
It is much more organized than the previous document.
Comments on the Quality of English LanguageI have no comment.
Author Response
Thank you
Reviewer 2 Report (New Reviewer)
Comments and Suggestions for Authors
General Comments: This study investigated the impact of sex and repetition maximum on kinematics and muscle activity during the sticking region of the barbell back squat. Results showed that women exhibited larger trunk lean and lower hip extension angles compared to men, potentially due to differences in hip/knee extensor strength ratios. Additionally, a shorter timing from V0 to Vmax2 was observed in lower repetition ranges, suggesting that repetition maximum may influence squat ascent requirements through variations in moments of inertia.
There is some text in red throughout the document which should be changed to black.
Avoid colloquial language in writing (see ‘grinding’ line 81)
Specific Comments
Introduction
-Lines 26-29: The first sentence needs a few citations to support points made.
-For the studies mentioned in the first paragraph (Larsen et al #8, 9, 10, 3) consider a brief descriptor of the training experience of the participants in the studies referred to.
-Lines 42-44: The first sentence of paragraph 2 can be improved. Please add citation(s) and also provide some examples of ‘athletic movements’ barbell load lifted has positive transfer to for athletes (e.g., jumping).
Line 74: the expression “cheaper oxygen demands” needs to be more clearly explained
-Line 84: What was the rationale for choosing 1, 3, 6 and 10 RM loads?
Methods
-How was the sample size determined? Please report details of the a priori power analysis if done. If not done please explain why.
-Please add more detail about what the powerlifting standards are for the readers do not need to find themselves if not familiar
-A diagram/figure to illustrate the methods would help readers to understand the procedures.
-What tempo were participants told to perform repetitions at? How much rest was allowed between reps? Did you provide any other instructions to the participants how to perform the repetitions?
-Line 119: I believe there is a typo and it should be 40%, not 4-
-Line 120: I understand the logic of randomizing the conditions in most cases; however, for this study if a subject performed a 1RM condition then 10RM next were you concerned that the 10RM performance could be altered due to post-activation potentiation (PAP)? Some training programs (e.g., French Contrast) would do something similar to take advantage of PAP.
-EMG paragraph (131-149): please refer to and cite EMG collection and processing standards that were followed. It appears that the SENIAM guidelines and recommendations by DeLuca et al. were adhered to, which is good but please cite.
-Statistics: were the data reviewed for outliers? Did you check for normality and other assumptions of 2-way RM ANOVA?
-For data in table 1 and 2 independent t-tests (or non-parametric equivalent) should be conducted and reported to confirm whether there were sex differences in these measures.
Results
-The first paragraph is confusing – what 50 m sprint are the authors referring to? Then there is reference to an equipment factor, laser gun? This appears to be results from a different study.
Line 230-232: please state you are referring to sEMG amplitude in the first sentence of this paragraph.
-Adding a few examples of individual kinematic vs. time and sEMG vs. time plots to the results would help to present the findings. A representative male and female data would be one approach. Also, based on the discussion of hip to knee extensor ratios a plot of trunk lean for a subject with low and high ratio would be beneficial for this key discussion point.
Discussion
-Paragraph 2: the hip to knee extensor points are good. Again adding some individual data traces to the results would help to illustrate the points.
-Lines 318-319: The durations noted here are important and accordingly reporting the tempo the participants lifted at, or other instructions provided, become important for comparing to prior studies.
-Additional limitations may need to be added pending responses to previous comments
Tables
Table 1 – units should be after the column labels: Height (cm)
Comments on the Quality of English LanguageSome editing the english throughout is warranted. The first paragraph of the results appears to be for another study.
Author Response
We want to thank the reviewer for the comments. We have answered to the comments of the reviewer and made the changes in the manuscript colored red. We think that it is suitable for publication now.
General Comments: This study investigated the impact of sex and repetition maximum on kinematics and muscle activity during the sticking region of the barbell back squat. Results showed that women exhibited larger trunk lean and lower hip extension angles compared to men, potentially due to differences in hip/knee extensor strength ratios. Additionally, a shorter timing from V0 to Vmax2 was observed in lower repetition ranges, suggesting that repetition maximum may influence squat ascent requirements through variations in moments of inertia.
There is some text in red throughout the document which should be changed to black.
The red text is, because it is a revision from the first version that we have send in. The red in the new version are the changes made according to the comments from you.
Avoid colloquial language in writing (see ‘grinding’ line 81)
Changed “grinding” to “lifting”
Specific Comments
Introduction
-Lines 26-29: The first sentence needs a few citations to support points made.
Added “McKean, M.R., P.K. Dunn, and B.J. Burkett, Quantifying the movement and the influence of load in the back squat exercise. The Journal of Strength & Conditioning Research, 2010. 24(6): p. 1671-1679”
-For the studies mentioned in the first paragraph (Larsen et al #8, 9, 10, 3) consider a brief descriptor of the training experience of the participants in the studies referred to.
Added “among strength-trained individuals” to the text now
-Lines 42-44: The first sentence of paragraph 2 can be improved. Please add citation(s) and also provide some examples of ‘athletic movements’ barbell load lifted has positive transfer to for athletes (e.g., jumping).
Added “(Stone, M., M. Stone, and H. Lamont, Explosive exercise. National Strength and Conditioning Association Journal, 1993. 15(4): p. 7-15)”
However, we don’t see the relevance of adding examples such as jumping etc. as this article is not about the transfer of maximal squat load into other performance-variables. The argument was made in the context of describing the major relevance of 1RM testing by its prevalence in multiple athletic cohorts
Line 74: the expression “cheaper oxygen demands” needs to be more clearly explained
Changed “cheaper” to “lower”
-Line 84: What was the rationale for choosing 1, 3, 6 and 10 RM loads?
The selection of 1, 3, 6, and 10 RM was based on a strategic rationale aimed at encompassing a broad spectrum of popular repetition ranges within resistance training. This range allows for a comprehensive analysis of neuromuscular responses and performance adaptations across low, moderate, and relatively high repetition levels. Additionally, this selection aligns with common practice in resistance training programs, which often include exercises performed at varying loadings to achieve diverse training outcomes, such as strength and hypertrophy.
Methods
-How was the sample size determined? Please report details of the a priori power analysis if done. If not done please explain why.
Unfortunately no priori power test was conducted
-Please add more detail about what the powerlifting standards are for the readers do not need to find themselves if not familiar
Added “, which requires the femur to be parallel to the floor at bottom position.”
-A diagram/figure to illustrate the methods would help readers to understand the procedures.
We have added a figure to illustrate the methods
Changed “figure 1” to “figure 2”, “figure 2” to “figure 3”, “figure 3” to “figure 4”, “figure 4” to “figure5
-What tempo were participants told to perform repetitions at? How much rest was allowed between reps? Did you provide any other instructions to the participants how to perform the repetitions?
Added “The subjects were not restricted in concentric/eccentric tempo and used their self-selected tempo, but were not allowed to remain at lockout for longer than 2 s.”
-Line 119: I believe there is a typo and it should be 40%, not 4-
Added “0”
-Line 120: I understand the logic of randomizing the conditions in most cases; however, for this study if a subject performed a 1RM condition then 10RM next were you concerned that the 10RM performance could be altered due to post-activation potentiation (PAP)? Some training programs (e.g., French Contrast) would do something similar to take advantage of PAP.
While acknowledging that no methodology is flawless and conceding the potential for performance modification due to increased PAP, we assert that randomization was crucial in enhancing the reliability of our results in this context. From our perspective, the omission of randomization could have introduced greater threats to the study's reliability compared to its inclusion. This consideration is particularly pertinent given the significant metabolic demands associated with the higher repetition sets. Consequently, we maintain that the methodological decision to randomize was instrumental in mitigating biases and ensuring the integrity of our findings.
-EMG paragraph (131-149): please refer to and cite EMG collection and processing standards that were followed. It appears that the SENIAM guidelines and recommendations by DeLuca et al. were adhered to, which is good but please cite.
We have added: Placement of the different electrodes were done according to SENIAM recommendations [31].
-Statistics: were the data reviewed for outliers? Did you check for normality and other assumptions of 2-way RM ANOVA?
Yes. We performed a Shapiro-Wilk test to test normal distribution. This is now mentioned in the text: Data was expressed as means and standard deviations (SD) per sex and normality of data was assessed and confirmed using the Shapiro-Wilk test.
-For data in table 1 and 2 independent t-tests (or non-parametric equivalent) should be conducted and reported to confirm whether there were sex differences in these measures.
That is correct. We have written this in the text now: Differences in anthropometrics and lifted load at the different repetition maximums between the sexes were assessed with independent samples t-tests. In the results we have mentioned: Women were significantly lighter, shorter, and had a lower absolute and relative strength across all repetition ranges (Table 1 and 2).
Results
-The first paragraph is confusing – what 50 m sprint are the authors referring to? Then there is reference to an equipment factor, laser gun? This appears to be results from a different study.
This is a big mistake, which is correct is from another study. We have deleted it now from the text.
Line 230-232: please state you are referring to sEMG amplitude in the first sentence of this paragraph.
Added “in sEMG amplitude”.
-Adding a few examples of individual kinematic vs. time and sEMG vs. time plots to the results would help to present the findings. A representative male and female data would be one approach. Also, based on the discussion of hip to knee extensor ratios a plot of trunk lean for a subject with low and high ratio would be beneficial for this key discussion point.
Unfortunately, we don’t have data on the hip/knee extensor data ratio. The discussion is made based on previous findings.
Discussion
-Paragraph 2: the hip to knee extensor points are good. Again adding some individual data traces to the results would help to illustrate the points.
Unfortunately, we don’t have data on the hip/knee extensor ratio. The discussion is made based on previous findings.
-Lines 318-319: The durations noted here are important and accordingly reporting the tempo the participants lifted at, or other instructions provided, become important for comparing to prior studies.
As we decided that the subjects could perform the lifts in their own tempo (ecological, we think most transferable to the real strength training situation) We can’t report the tempo of the subjects. Furthermore, this is also not a main point of our study. We were only interested if there are differences in kinematics in the last rep at different RMs and if this was different between sexes. We think that everybody understands that 10-RM takes longer time than 1-RM and that this is not interesting to also calculate for all subjects.
-Additional limitations may need to be added pending responses to previous comments
We have added to the text: Thirdly, lifting tempo was not controlled for, which could influence the results. However, to avoid extra constraints on the subjects, which would make this study less ecological, it was decided that lifting tempo and barbell placement between subjects was not standardised.
Tables
Table 1 – units should be after the column labels: Height (cm)
Added “(years), “(cm)”, “(kg)”
Comments on the Quality of English Language
Some editing the english throughout is warranted. The first paragraph of the results appears to be for another study.
We have gone through the whole manuscript and changed it where necessary.
Round 2
Reviewer 2 Report (New Reviewer)
Comments and Suggestions for Authors
I am pleased with most of the changes the authors have made. There does appear to be some confusion a previous comment regarding the results which was not addressed.
The comment:
"Adding a few examples of individual kinematic vs. time and sEMG vs. time plots to the results would help to present the findings. A representative male and female data would be one approach. Also, based on the discussion of hip to knee extensor ratios a plot of trunk lean for a subject with low and high ratio would be beneficial for this key discussion point."
I am asking you to add plots of YOUR data for 1) kinematic vs. time 2) emg vs. time and 3) trunk lean vs. time
Author Response
I am pleased with most of the changes the authors have made. There does appear to be some confusion a previous comment regarding the results which was not addressed.
Thank you
The comment:
"Adding a few examples of individual kinematic vs. time and sEMG vs. time plots to the results would help to present the findings. A representative male and female data would be one approach. Also, based on the discussion of hip to knee extensor ratios a plot of trunk lean for a subject with low and high ratio would be beneficial for this key discussion point."
I am asking you to add plots of YOUR data for 1) kinematic vs. time 2) emg vs. time and 3) trunk lean vs. time
We have included a plots of joint kinematics over time for a representative man and woman in the discussion to show how the development is. About EMG over time we think that a graph of EMG over time for the muscles do not add extra information to the manuscript as all information is already shown around the sticking region in figure 5.
We hope that the reviewer appreciates our point of view about.
This manuscript is a resubmission of an earlier submission. The following is a list of the peer review reports and author responses from that submission.
Round 1
Reviewer 1 Report
Comments and Suggestions for Authors
This study aimed to determine the possible differences in squat mechanics (kinematics and sEMG) between the sexes and at different repetition maximums (1, 4, 6, 10).
Although this study could be of interest, the writeup (manuscript) has many flaws and cannot be readily reviewed. Firstly, the Introduction section abruptly ends with no purpose or hypothesis statements and thus the reader is left with no real idea what is to be found in the Methods. Second, the Methods are not only lacking important information (you have no data reduction/analysis section regarding your EMG or kinematic data), it also suggests issues in your study (e.g., not accounting for differences in lifting shoes - this is a well known issue in biomechanics). Third, your statistical assessment is flawed. You have an alpha level that is much too lax given you are testing many variables, your model isn't "two-way with repeated measures" it is a mixed model with a group and repeated measures, and your approach is problematic. Your statistical assessments should match your purpose (which doesn't exist). As it is currently written, you really couldn't compare your kinematics and EMG as they exist at different ranges of time. Differences in joint angles at two different instances may have no real link at all with the average sEMG; yet, presumably, your plan was to combine these data to tell a story.
Citation #1 is unnecessary and inappropriate. This study did not determine the applications of back squats in the way this is proposed herein.
Citation #12 is also used inappropriately. I am unsure why you need a citation to NSCA (whom did not originate 1RM testing) to state that 1RM performance is indicative of your ability to exert force – that is what the 1RM is.
“In addition, women possess a knee/hip 69 strength ratio close to 1, suggesting equal strength in knee and hip extensors”
This statement requires citation support.
“Moving on,…”
This is odd.
“To mitigate the risk of a learning effect, each participant underwent a familiarisation test >72 hours before the experimental test session”
What learning effect can be obtained when these are trained males and females?
“The use of lifting aids (e.g., knee sleeves, lifting belt) was 107 prohibited, with the exception of lifting shoes”
Lifting shoes can make a huge difference – this is a well known fact in biomechanics. You need to modify your assessments based on who used these shoes.
“To minimise the risk of fatigue-induced performance, each participant was required to rest for a minimum of five minutes between each set for both test sessions.”
5 minutes quite long. The standards are up to 5 minutes, not at least 5 minutes.
“A linear encoder sampling at 200 Hz (ET-Enc-02, Ergotest Technology AS, Langesund, Norway) was used to synchronise sEMG data to velocities and thereby sample sEMG data for the pre-sticking, sticking, and post-sticking regions”
To what velocities? Until now you’ve not stated anything regarding velocity.
“This facilitated the determination of sagittal-plane kinematics for the hip, knee, and ankle joints, which in an erect standing position were defined as 180° for hip and knee, and 90° for the ankle. Trunk lean angle was defined as the torso relative to the laboratory, with the erect position defined as 0°”
Why are these different. Why correct trunk lean to 0 at erect but leave joint angles at 180 deg an 90 deg?
Reviewer 2 Report
Comments and Suggestions for Authors
Thank you for submitting to JFMK.
The value of a study is important whether it is significant or not significant as an experimental result. But more importantly, how will these results be applied in the field?
1. The sample size is too small to represent men and women. (What is the basis for calculating the sample size?)
2. In the author's research, what is more important than simply information about differences between men and women is the usability of the information. Electromyography experiments must be connected to the following two things. The first is performance or training science and the second is injury-related causes, mechanisms and rehabilitation implications. This content is too limited in the introduction and discussion.
3. There is too little participant information to understand this study. General information such as age in the abstract is missing from the text. The research method in the text requires a table that can confirm the characteristics of the participants.
4. Crucially, it is not easy to interpret because there are too many bar graphs. Research is very important in resolving the author's intellectual curiosity. However, publishing requires you to think about your readers. authors expressed RM 1, 3, 6, and 10. And then V0 Vmin Vmax V2. Even though authors have set up so many variables, it seems very difficult for the reader to understand so many graphs. In the case of figures 1 and 4, various graphs are shown. Classify (A), (B), (C), and (D) and explain.
Comments on the Quality of English LanguageMore scientific and academic expressions are needed.